# Thinking is Seeing: Multi-modal Large Language Models are Exceptional in Understanding Knowledge Graphs

## Abstract

The representation learning of knowledge graphs (KGs) is a longstanding research problem. While graph neural networks (GNNs) have driven recent progress, they still struggle with encoding textual features and subtle relationships of KGs, particularly in conveying key information to large language models (LLMs). The emergence of multi-modal LLMs (MLLMs), which combine linguistic and visual understanding, presents an intriguing opportunity: Could their vision capabilities inspire mental visualization, facilitating conceptual thinking and abstract reasoning akin to human cognition? To investigate this premise, we propose SeeKG, an innovative framework that transforms KGs into visually rendered representations as image inputs for MLLMs. We evaluate SeeKG under both training-free and supervised fine-tuning settings, where the experimental results show that SeeKG excels in understanding KG sub-graphs and achieves competitive performance even without training or demonstrations. Further fine-tuning on small-batch data reveals that it outperforms state-of-the-art LLM-based KG completion methods by substantial margins across multiple benchmark datasets.

## 1 Introduction

Knowledge graphs (KGs) store real-world facts in the form of triples and serve as fundamental resources for many AI applications (Ji et al., 2020). With the rise of deep neural models, learning the low-dimensional vectors (i.e., embeddings) for KG entities and relations gains increasingly more attention (Bordes et al., 2013; Trouillon et al., 2016; Sun et al., 2019; Guo et al., 2019; Zhu et al., 2021). In recent years, graph neural networks (GNNs) (Kipf & Welling, 2017) have become the standard for state-of-the-art KG representation learning. They take the adjacency matrices as input and stack multiple GNN layers to aggregate near-to-far neighborhood information of the given entity (Nguyen et al., 2018; Vashishth et al., 2020; Chen et al., 2021; Cui et al., 2024).

However, the resulting embeddings often fail to fully capture contextual information. While the GNN-based methods efficiently summarizes variable-length relationships into a single vector, the detailed information of neighboring entities is discarded and the whole process is irreversible. Such deficiencies become more severe in integrating KGs with large language models (LLMs), where fine-tuning is compulsory for aligning structural KG embeddings with LLM token embeddings. Even so, the LLMs still struggle to interpret the basic properties of the encoded information (e.g., identifying entities in the contextual KG) (Lin et al., 2025; Zhang et al., 2025).

Take the input KG image in Figure 1 as an example, many conventional KG representation learning methods ignore the textual information of entities (e.g., the name of "Bulat Okudzhava"), or merely aggregate word embeddings as initial features via bag-of-word models (Chen et al., 2022; Zhang et al., 2024a). Consequently, the LLM is incapable of inferring the detailed contextual KG information (e.g., whether *Moscow* exists in the KG) from the input embeddings, as they mix a varying number of neighboring entities in an unordered manner. Integrating these structural encoders into LLMs remains highly challenging even fine-tuning on the full training set (Yao et al., 2023; Zhang et al., 2024b; Guo et al., 2024a; 2025; Lin et al., 2025).

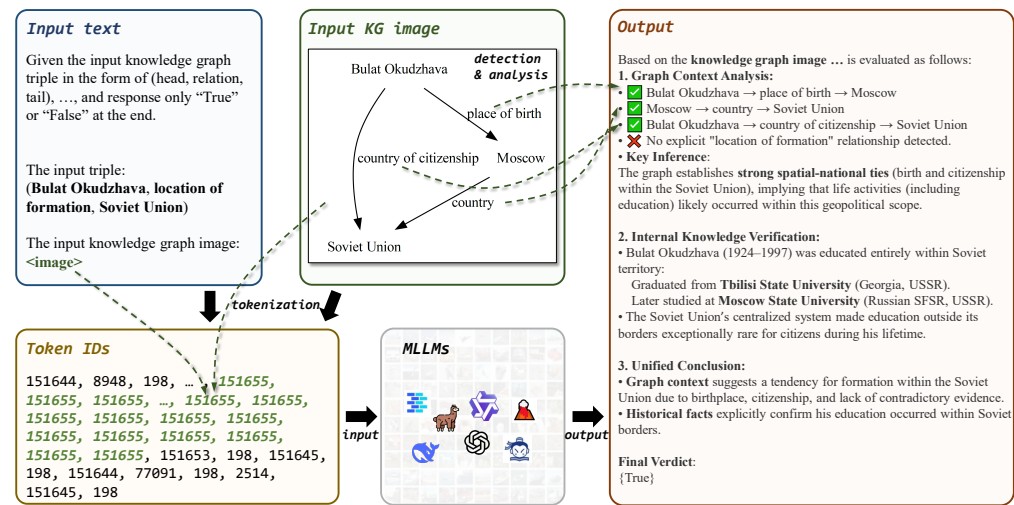

Figure 1: The workflow of SeeKG. The input text and KG image are first tokenized into IDs by the tokenizer, where the image is sampled from the CoDeX-S (Safavi & Koutra, 2020) dataset and generated by our visualization tool. During tokenization, the special tag <image> is converted to image paddings, which are then replaced by image token embeddings when fed into the MLLM. The output demonstrates that the MLLM sufficiently understands the graph context in the KG image and integrates the visual clues with its internal knowledge to correctly predict the answer.

KGs are highly abstract graphs, yet humans intuitively understand and reason about them through visualization, i.e., sketching them manually or imaging them mentally. In some sense, thinking is seeing (Danesi, 1990; Arnheim, 2023). Abstract thinking can be linked to the visual system.

Multi-modal large language models (MLLMs) facilitate LLMs with the vision capability to perceive and interact with the world (Achiam et al., 2023; Lin et al., 2024; Wu et al., 2025). Technically, prevalent convolution operations and local attention mechanisms (e.g., window-based attention (Liu et al., 2021)) are capable of capturing sub-graph information in the KG image, enabling better structural comprehension compared to processing textualized KG triple sequences.

With new MLLMs emerging rapidly, it is interesting to query whether the vision understanding of MLLMs can surpass conventional KG representation learning in relevant tasks? Can they interpret and complete KGs by inferring the visualized images like humans?

To answer the above questions, we propose *SeeKG*, an end-to-end framework to extract, visualize, and reason about KG subgraphs with MLLMs. Figure 1 illustrates an example of how SeeKG works. The visualized KG image is processed through the vision encoder in the MLLM and then mapped to the special token <image> in the input instruction. The subsequent steps mirror the standard LLM reasoning, where the MLLM takes the token embedding sequence as input and generates the analysis and answer accordingly.

We summarize our contributions as follows:

- We explore a novel direction of leveraging visual modality to represent KG structural information, investigating the extent to which MLLMs can comprehend and reason over the abstract KG images.
- We propose SeeKG, an end-to-end framework for visualizing real-world KGs as images and reasoning with MLLMs. SeeKG supports diverse MLLMs, sub-KG sampling strategies, and customizable visualization settings.
- We evaluate SeeKG on triple classification task, which is one of the most important tasks for KG completion. The experimental results across multiple datasets show that SeeKG significantly outperforms all LLM-based methods in training-free setting and achieves state-of-the-art performance with supervised fine-tuning.

## 2 RELATED WORKS

**Multi-modal Large Language Models**  Understanding and generating multi-modal content has become fundamental for advanced artificial intelligence systems. Consequently, MLLM research has attracted significant attention across communities ranging from natural language processing (NLP) and computer vision to audio processing and multi-modal learning (Yu et al., 2021; Lin et al., 2024; Abdin et al., 2024; Hurst et al., 2024; Wu et al., 2025). A variety of MLLM families are established, such as LLaVa (Lin et al., 2024), Qwen-VL (Wu et al., 2025), DeepSeek-VL (Lu et al., 2024), MiniCPM (Aharoni & Goldberg, 2020), InternVL (Chen et al., 2024), GPT-o (Hurst et al., 2024), and Gemini (Comanici et al., 2025). These models demonstrate versatile capabilities, processing not only conventional photographs but also structured content like tables, code snippets, and mathematical expressions, all in the image form. Building on these advances, we propose SeeKG to harness MLLMs for knowledge graph reasoning.

**Large Language Models for Knowledge Graphs**  With the rapid advancement of LLMs, there are increasingly more works that integrate them into KG tasks (Yang et al., 2023; Pan et al., 2024; 2023; Guan et al., 2024; Yao et al., 2023; Zhang et al., 2024b; Guo et al., 2024a; 2025; Lin et al., 2025). The most straightforward approach is employing prompt engineering to convert triples from structural ID tuples into textual short sentences (Yao et al., 2023). Contextual information like neighboring entities can be also textualized as LLM input (Zhang et al., 2024b). However, such graph-to-text conversion risks losing critical structural information, resulting in suboptimal task performance (Zhang et al., 2024b; Guo et al., 2024a).

To enhance the structural understanding, recent methods (Zhang et al., 2024b; Guo et al., 2024a; Lin et al., 2025) leverage conventional KG representation learning models as structural encoders, aligning their output embeddings with LLMs through supervised fine-tuning. For example, KoPA (Zhang et al., 2024b) employs the triple-based models RotatE (Sun et al., 2019) to obtain the embeddings of entities and relations, while MKGL (Guo et al., 2024a) employs PNA (Corso et al., 2020) as graph encoder to encode structural information. SSQR (Lin et al., 2025) learns to map the structural information of KG to special code token embeddings via quantized representations (Kostant, 2006).

These approaches train adapters to bridge structural encoders and LLMs, resembling the training process of many MLLMs. However, the KG representation learning methods mainly capture the high-level structural information. Even those text-aware variants (Lee et al., 2023; Zhang et al., 2024a) lose identifiable details through the average-based aggregation of GNNs. It is still challenging to enabling LLMs to understand the contextual output of the structural encoders.

In this paper, we investigate a novel direction: rather than adapting MLLMs for multi-modal KG tasks, we explore the correlation between the visual capacity of MLLMs with abstract thinking.

## 3 METHODOLOGY

In this section, we present the details of proposed SeeKG, discussing and analyzing how to employ an MLLM for KG reasoning. We begin with preliminaries that introduce the fundamental concepts of KGs and MLLMs, then demonstrate how we construct KG images as contextual input for MLLMs. Finally, we illustrate the two different implementations of SeeKG.

### 3.1 PRELIMINARIES

**Knowledge Graphs**  We define a KG as $\mathcal{G} = \{\mathcal{T}, \mathcal{E}, \mathcal{R}\}$, where $\mathcal{T}$, $\mathcal{E}$, $\mathcal{R}$ represent the sets of triples, entities, and relations, respectively. Each triple $\tau = (e_i, r_k, e_j) \in \mathcal{T}$ encodes a factual relationship where entity $e_i \in \mathcal{E}$ relates to entity $e_j \in \mathcal{E}$ through relation $r_k \in \mathcal{R}$. It is worth noting that, conventional KG representation learning methods typically focus solely on the structure while disregarding multi-modal information, organizing inputs as raw IDs, e.g., $\tau = (0, 2, 1)$ where $e_i = 0$, $e_j = 1$, and $r_k = 2$.

**Multi-modal Large Language Models**  MLLMs are LLMs capable of processing and generating multi-modal features, such as images and videos. Most MLLMs leverage pretrained LLMs as their foundation, aligning visual (and other modality) embeddings with the token space of original LLMs

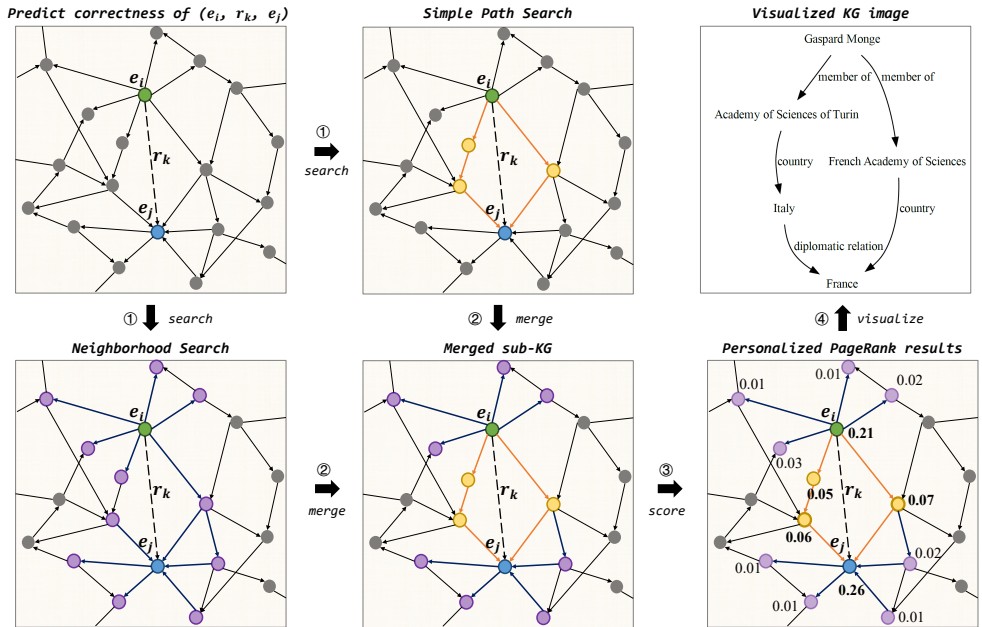

Figure 2: The pipeline for constructing contextual KG images. (1) we consider simple paths as the key contextual sub-KG, with multi-hop neighboring entities as complement; (2) all obtained paths and neighboring entities are merged into a unified sub-KG; (3) we leverage Personalized PageRank (PPR) (Gasteiger et al., 2018) to compute the importance scores of the entities in the merged sub-KG, where the source entity $e_1$ and target entity $e_2$ are initialized with weights of 1, and others set to 0; (4) we visualize the structural sub-KG as an KG image using Graphviz (Ellson et al., 2004).

through post-training techniques (Abdin et al., 2024; Lin et al., 2024). We formalize an MLLM as $\mathcal{M}$, which takes an instruction text $t_{in}$ and image $v_{in}$ as input to generate a textual response $t_{out}$, i.e., $t_{out} = \mathcal{M}(t_{in}, v_{in})$.

**LLM for KG Completion**  KG completion is one of the most crucial tasks for KG representation learning. It encompasses multiple sub-tasks, including entity linking (Bordes et al., 2013), relation prediction (Yao et al., 2019), triple classification (Safavi & Koutra, 2020), etc. In this paper, we focus on triple classification, a task analogous to conventional question answering that has gained prominence in evaluating LLM-based KG reasoning methods (Yao et al., 2023; Zhang et al., 2024b; Lin et al., 2025). Specifically, given a triple $\tau = (e_i, r_k, e_j)$, the LLM is required to determine the validity of this triple and output either "True" or "False" as the final response.

To leverage the internal knowledge and enhance the comprehension of LLMs towards the given triple $\tau$, the input IDs $(e_i, r_k, e_j)$ (also the relevant contextual triples) is transformed into textual format, such as *(Bulat Okudzhava, place of birth, Moscow)*. With appropriate prompts, the LLMs inherently demonstrates promising zero-shot prediction capability for triple classification.

## 3.2  KNOWLEDGE GRAPHS AS IMAGES

We illustrate the workflow of generating a contextual sub-KG visualization for the given triple $\tau$ in Figure 2. As our goal is to extract the most informative sub-KG as context, the primary objective is to collect the potentially helpful entities and triples surrounding $\tau$.

Drawing upon the established approaches in KG representation learning, we observe that both path-based and neighborhood-based contexts have consistently served as fundamental elements for reasoning systems (Lin et al., 2015; Guo et al., 2019; Vashishth et al., 2020; Chen et al., 2021; Guo et al., 2024a; Lin et al., 2025). This empirical evidence inspires us to integrate these two contextual dimensions when building the sub-KG structure for $\tau$.

**Simple Path Search**   While path discovery between KG entities presents computational challenges due to potential instability and time complexity, identified paths carry significant semantic value. The presence of interconnected paths strongly indicates entity correlations and provides interpretable evidence for verifying the validity of the given triple.

To efficiently collect the path context information, we employ *simple path search* (Goldberg & Harrelson, 2005) on the KG, and designate $e_1$ as the source entity and $e_2$ as the target entity for $\tau = (e_i, r_k, e_j)$. A *simple path* $p$ is a path in the KG that does not have repeating entities. We enforce the simple path constraint to prohibit entity repetitions within any single path, ensuring computational tractability and eliminating redundant entities:

$$p = \{(e_n, r_n, e_{n+1})\}_{n=1}^{l(p)}, \quad \text{where } e_n \neq e_m, \forall n \neq m \tag{1}$$
$$\mathcal{P}_{e_i \to e_j} = \{p | s(p) = e_i \,\&\, t(p) = e_j \,\&\, l(p) \leq L\} \tag{2}$$

where $p$ represents a simple path and $\mathcal{P}_{e_i \to e_j}$ denotes the complete set of valid paths from $e_i$ to $e_j$. $l(\cdot)$ is the path length function. We set a hyperparameter $L$ to balance exploration depth and computational efficiency. $s(\cdot)$ and $t(\cdot)$ identify path origins and destinations, respectively. With the resulting simple path set $\mathcal{P}$, we can construct a contextual sub-KG $\mathcal{G}_{e_i \to e_k}$, where path edges become subgraph relations and traversed entities form the node set.

**Neighborhood Search**   When no paths exists or only a limited number of paths are available, we may augment the context sub-KG by incorporating the $K$-hop neighbors of the source entity $e_i$ and target entity $e_k$. Formally, the neighboring entity set $\mathcal{E}_{e_i, e_k}$ is defined as:

$$\mathcal{E}_{e_i, e_k} = \{e | d(e, e_i) \leq K\} \cup \{e | d(e, e_k) \leq K\}, \tag{3}$$

where $d(\cdot, \cdot)$ measures the distance between a pair of entities. $K$ denotes the maximal distance threshold to bound the search scope. Subsequently, we can construct a sub-knowledge graph $\mathcal{G}_{e_i, e_k}$ with entities in $\mathcal{E}_{e_i, e_k}$ as nodes.

**Knowledge Graph Pruning**   Although the path sub-KG $\mathcal{G}_{e_i \to e_k}$ and neighborhood sub-KG $\mathcal{G}_{e_i, e_k}$ offer essential contextual information for triple validation, merging them into a unified sub-KG $\mathcal{G}_\tau$ may also introduce substantial redundancy, especially when $e_i$ and $e_j$ exhibit dense connectivity. In this case, the information overload may exceed the processing capacity of LLMs.

To mitigate this problem, we leverage Personalized PageRank (PPR) (Page et al., 1999; Gasteiger et al., 2018) to prune $\mathcal{G}_\tau$. Specifically, we initialize the weights of the source entity $e_i$ and target entity $e_j$ as 1 while setting others to 0. Then, we apply PPR on $\mathcal{G}_\tau$ to compute the importance scores of all involved entities:

$$\mathbf{w}_{\mathcal{E}_\tau}^{out} = \text{PPR}(\mathcal{G}_\tau, \mathbf{w}_{\mathcal{E}_\tau}^{in}), \tag{4}$$

where $\mathbf{w}_{\mathcal{E}_\tau}^{out}$ is output score vector for entities in $\mathcal{E}_\tau$, while $\mathbf{w}_{\mathcal{E}_\tau}^{in}$ represents the initial input with $e_i = 1$ and $e_j = 1$ (others 0). Only entities with top-$N_e$ scores are retained in the final sub-KG $\mathcal{G}_\tau$.

PPR has significant advantages over the existing embedding-based or LLM-based approaches in computational speed. Meanwhile, it also precisely meets our intention: when multiple paths exist between the source entity $e_i$ and target entity $e_j$, entities on these paths receive higher scores due to their connectivity importance. If no direct path exists, the entities with high scores exhibit significant betweenness centrality or proximity to $e_i$ and $e_j$, avoiding being erroneously pruned.

**Visualization**   While numerous open-source tools (e.g., Neo4j visualization library, yFiles, ipysigma, etc.) can be used for visualizing the sub-KGs, most of them prioritize user interaction and algorithmic features, which diverge from our focus on enabling rich stylistic customization for $\mathcal{G}_\tau$. Therefore, we develop a dedicated visualization tool based on Graphviz (Ellson et al., 2004), a versatile framework for graph rendering.

For the main experiments, entity names in the visualized images are displayed in *Times New Roman* without decorative elements (e.g., boxes or circles) to enhance clarity. Relation labels are placed directly around edges. Thus, the generated KG images are friendly for optical character recognition (OCR) and readable for both humanity and machine.

Table 1: The main results on triple classification. "-" denotes the unavailable entries. The best and second-best Acc/F1 results are **boldfaced** and underlined, respectively.

| | Model | CoDeX-S | | | | FB15K-237N | | | |
|---|---|---|---|---|---|---|---|---|---|
| | | Acc | P | R | F1 | Acc | P | R | F1 |
| Embedding-based | TransE (Bordes et al., 2013) | 72.07 | 71.91 | 72.42 | 72.17 | 69.71 | 70.80 | 67.11 | 68.91 |
| | DistMult (Yang et al., 2015) | 66.79 | 69.67 | 59.46 | 64.16 | 58.66 | 58.98 | 56.84 | 57.90 |
| | ComplEx (Trouillon et al., 2016) | 67.64 | 67.84 | 67.06 | 67.45 | 65.70 | 66.46 | 63.38 | 64.88 |
| | RotatE (Sun et al., 2019) | 75.68 | 75.66 | 75.71 | 75.69 | 68.46 | 69.24 | 66.41 | 67.80 |
| LLM-based Training-free | Zero-shot (Alpaca) (Taori et al., 2023) | 50.62 | 50.31 | 99.83 | 66.91 | 56.06 | 53.32 | 97.37 | 68.91 |
| | Zero-shot (GPT-3.5) (Achiam et al., 2023) | 54.68 | 69.13 | 16.94 | 27.21 | 60.15 | 86.62 | 24.01 | 37.59 |
| | ICL (1-shot) (Zhang et al., 2024b) | 49.86 | 49.86 | 50.59 | 50.17 | 54.54 | 53.67 | 66.35 | 59.34 |
| | ICL (2-shot) | 52.95 | 51.54 | 98.85 | 67.75 | 57.81 | 56.22 | 70.56 | 62.58 |
| | ICL (4-shot) | 51.14 | 50.58 | 99.83 | 67.14 | 59.29 | 57.49 | 71.37 | 63.68 |
| | ICL (8-shot) | 50.62 | 50.31 | 99.83 | 66.91 | 59.23 | 57.23 | 73.02 | 64.17 |
| | SeeKG | **72.78** | 69.66 | 80.74 | **74.79** | **67.92** | 64.14 | 81.32 | **71.71** |
| LLM-based Fine-tuning | KG-LLaMA (Yao et al., 2023) | 79.43 | 78.67 | 80.74 | 79.69 | 74.81 | 67.37 | 96.23 | 79.25 |
| | KG-Alpaca (Yao et al., 2023) | 80.25 | 79.38 | 81.73 | 80.54 | 69.91 | 62.71 | 98.28 | 76.56 |
| | Vanilla IT (Zhang et al., 2024b) | 81.18 | 77.01 | 88.89 | 82.52 | 73.50 | 65.87 | 97.53 | 78.63 |
| | Structure-aware IT (Zhang et al., 2024b) | 81.27 | 77.14 | 88.40 | 82.58 | 76.42 | 69.56 | 93.95 | 79.94 |
| | KoPA (Zhang et al., 2024b) | 82.74 | 77.91 | 91.41 | 84.11 | 77.65 | 70.81 | 94.09 | 80.81 |
| | SSQR (Lin et al., 2025) | - | - | - | - | 79.80 | 75.90 | 87.20 | 81.10 |
| | SeeKG | **84.05** | 80.44 | 89.99 | **84.95** | **82.20** | 80.18 | 85.55 | **82.78** |

## 3.3 REASONING KG IMAGES WITH MLLMS

Now, we are capable of employing an MLLM for triple classification with KG images as context. Following the existing works that leverage LLMs for KG completion, we implement SeeKG through two prevalent strategies:

**Training-free SeeKG** The training-free strategy directly leverage pretrained LLMs for question answering, where prompt design is critical. The recent studies demonstrate significant improvements through techniques like in-context-learning (ICL) (Dong et al., 2022; Wies et al., 2024) and chain-of-thought (CoT) (Wei et al., 2022). However, we deliberately adopt a minimalistic prompting approach for fair comparison with baseline methods in the main experiment, which is sufficient to surpass them by substantial margins. The instruction template is presented in Appendix B, in which the MLLMs are required to response strictly with "True" or "False" only.

**Supervised Fine-tuning SeeKG** Our designed visualization style have simplified the sub-KG representation, but the resulting KG images remain unfamiliar to MLLMs. Therefore, performing supervised fine-tuning can enhance the understanding of MLLMs towards the triple classification task, thereby minimizing the knowledge gap. We follow the existing LLM-based methods (Zhang et al., 2024b) to perform supervised fine-tuning to optimize SeeKG with identical instruction template used in the training-free setting.

## 4 EXPERIMENT

In this section, we conduct a series of experiments to verify the effectiveness of SeeKG. The source code has been uploaded and will be available on GitHub.

## 4.1 SETTINGS

**Datasets** We consider CoDeX-S (Safavi & Koutra, 2020) and FB15K-237N (Lv et al., 2022) as datasets, which are widely used in evaluating LLM-based methods for triple classification. The detailed statistics of these two datasets can be found in Appendix C.

**Implementation** We use Qwen2.5-VL-7B (Wu et al., 2025) as the backbone MLLM for SeeKG in the main experiment. Qwen2.5-VL is one of the most popular MLLMs in AI community, and the parameter size of its 7B version is analogous to that of the LLMs employed in baselines. In the supervised fine-tuning setting, we employ low-rank adaption (LoRA) (Hu et al., 2021) for efficiency,

Table 2: Ablation studies under supervised fine-tuning setting on CoDex-S and FB15K-237N.

| Model | CoDeX-S | | | | FB15K-237N | | | |
|---|---|---|---|---|---|---|---|---|
| | Acc | P | R | F1 | Acc | P | R | F1 |
| SeeKG | **84.05** | 80.44 | 89.99 | **84.95** | **82.20** | 80.18 | 85.55 | **82.78** |
| - w/o context | 81.40 | 76.92 | 89.72 | 82.83 | 74.26 | 67.36 | 94.13 | 78.53 |
| - w/ text context | 82.71 | 83.37 | 81.73 | 82.54 | 77.90 | 70.85 | 94.80 | 81.09 |
| - w/ image + text context | _83.37_ | 79.13 | 90.65 | _84.50_ | _81.88_ | 83.67 | 79.22 | 81.39 |
| - w/o simple paths | 82.71 | 83.37 | 81.73 | 82.54 | 80.72 | 78.12 | 85.34 | 81.57 |
| - w/o neighborhood | 83.34 | 81.66 | 86.00 | 83.77 | 81.48 | 78.42 | 86.87 | _82.43_ |
| - w/ random pruning | 81.62 | 78.47 | 87.14 | 82.58 | 79.65 | 77.68 | 83.21 | 80.35 |

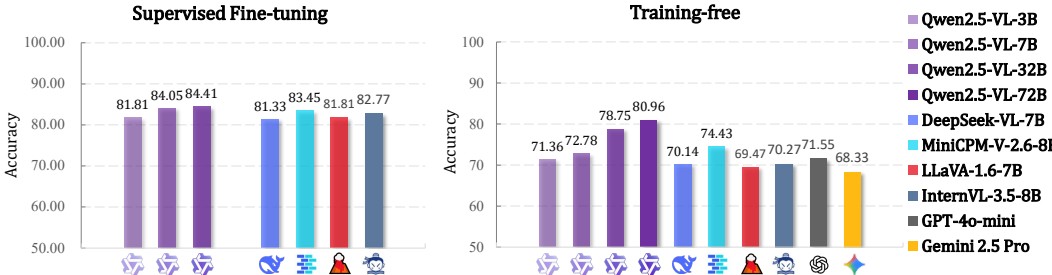

Figure 3: The accuracy results of SeeKG with different backbone MLLMs on the CoDeX-S dataset.

where $r = 64$ is identical to the baseline method KoPA (Zhang et al., 2024b). The learning rate and number of training epoch are set to $0.0001$ and $2.0$, respectively. The original training sets for supervised fine-tuning have over $100,000$ samples, while we find that only using $10,000 \sim 20,000$ samples is sufficient for fine-tuning SeeKG. The detailed settings can be found in Appendix D.1.

**Baselines and Metrics**  We compare our method with state-of-the-art baselines including: the conventional embedding-based methods, such as TransE (Bordes et al., 2013), DistMult (Yang et al., 2015), ComplEx (Trouillon et al., 2016), and RotatE (Sun et al., 2019); the LLM-based training-free methods, such as zero-shot and ICL-based methods (Zhang et al., 2024b); the LLM-based fine-tuning methods, such as KG-LLama (Yao et al., 2023), KG-Alpaca (Yao et al., 2023), KoPA (Zhang et al., 2024b), and SSQR (Lin et al., 2025). Following the existing works, we use accuracy (Acc), precision (P), recall (R), and F1-score (F1) as the evaluation metrics.

## 4.2 MAIN RESULTS

Table 1 presents the main experimental results on triple classification. Notably, the LLM-based fine-tuning methods substantially outperform conventional embedding-based approaches, demonstrating the superior capability of LLM on reasoning KGs. Within the LLM-based methods, SeeKG consistently achieves state-of-the-art performance in both training-free and supervised fine-tuning settings across the main metrics and datasets. These results provide empirical evidence that MLLMs excel at understanding KG images and generating reliable triple validity predictions.

Specifically, the LLM-based training-free baselines show no significant advantage over conventional embedding-based methods, which may be attributed to their unfamiliarity with the triple classification task. On the CoDeX-S dataset, anomalous recall patterns emerge in several LLM-based methods. These extreme values suggest model bias toward blanket acceptance/rejection of most triples. Results on FB15K-237N exhibit more balanced precision-recall tradeoffs, though accuracy remains below the best conventional method. It is worth noting that our SeeKG still achieves competitive or even better results, demonstrating robust superiority over the existing LLM-based methods.

For the fine-tuning setting, all LLM-based methods exhibit dramatic performance gains, decisively surpassing conventional methods. Our SeeKG emerges as new state-of-the-art across both datasets and primary metrics. Remarkably, SeeKG achieves these results with significantly reduced training data requirements, highlighting its effectiveness and efficiency.

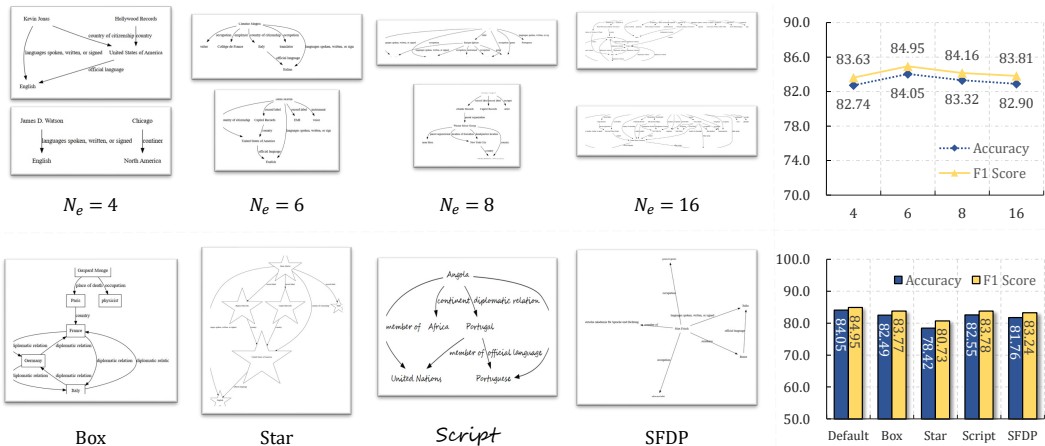

Figure 4: The results of SeeKG with different pruning and visualization settings on the CoDeX-S dataset. $N_e$ denotes the maximal number of entities. Box and Star are visualization styles featuring entities with rectangular and star-shaped borders, respectively. Script and SFDP denote style with cursive font and scaled force-directed placement layout, respectively.

## 4.3 ABLATION STUDY

We perform ablation studies on the CoDex-S and FB15K-237N datasets to validate the efficacy of each module in SeeKG. We develop several variants for comparison: "w/o context", "w/ text context" and "w/ image + text context" denote our method without sub-KG context, with textualized context, and with both two types of context, respectively. "w/o simple paths", "w/o neighborhood", and "w/ random pruning" are the methods excluding simple paths, omitting multi-hop neighborhood, and replacing PPR-based pruning with random pruning, respectively.

The results are shown in Table 2, from which we can observe that: The proposed method outperforms all variants across two datasets. The closest competitor is SeeKG with image + text context, though its performance remains slightly inferior. Given that modern MLLMs excel at OCR, the triple information loss in KG images is nearly negligible, especially in the fine-tuning setting. In contrast, textual context may divert the attention of the MLLM away from KG images, which explicitly illustrate structural correlations among entities and relations.

For sub-KG construction, removing simple paths results in significantly more performance degradation compared to removing multi-hop neighborhood information, implying that paths serve as more critical context for KG reasoning. PPR pruning is also vital for sampling high-quality sub-KGs, as replacing it with random pruning also causes performance drop on both datasets.

## 4.4 BACKBONE MLLMs

We evaluate SeeKG with various backbone MLLMs under both training-free and supervised fine-tuning settings, and present results on the CoDeX-S dataset in Figure 3. Overall, larger MLLMs generally achieve better performance in both settings. Qwen2.5-VL-32B and Qwen2.5-VL-72B emerge as the top-performing models, with accuracy results in training-free setting approaching those of fine-tuned 7B models. The minimal performance gap between these two methods may stems from that the 32B model is newly released with reinforcement learning training.

The other open-source MLLMs also achieve promising results, especially MiniCPM-V-2.6-8B, which significantly outperforms Qwen2.5-VL-7B in the training-free setting. The Commercial MLLMs GPT-4o-mini and Gemini 2.5 Pro underperform relative to our expectations, despite repeated adjustments to the instructions. The primary reason appears to be their overly conservative decision-making, resulting in very low recall scores. Nevertheless, most MLLMs still outperform the single-modal LLMs presented in Table 1. More detailed results can be found in Appendix D.2.

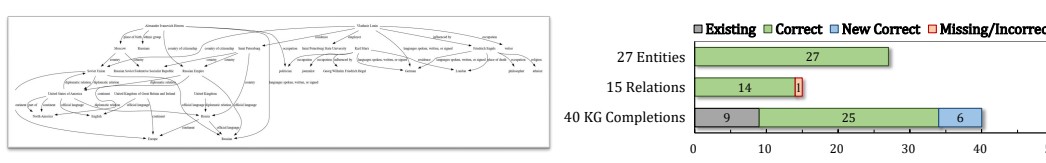

a. A complicated KG image contains 27 entities and 41 triples.    b. Results of Qwen2.5-VL-72B for three tasks

Figure 5: The case study results. We present the MLLM with a complex KG image and evaluate its performance on three tasks: listing all entities, listing all relations, and completing 40 triples.

### 4.5 PRUNING AND VISUALIZATION SETTINGS

We conduct experiments to further analyze the effects of different pruning and visualization settings. As shown in Figure 4, the top sub-figures present sampled KG images with varying maximal entity number $N_e$ and their corresponding performance on the CodeX-S dataset. SeeKG maintains robust performance across different $N_e$ values, consistently achieving state-of-the-art results. The optimal configuration emerges at $N_e = 6$, where the corresponding KG images have more appropriate node/edge density. When $N_e = 4$, the KG images often fail to capture essential paths between source and target entities, causing noticeable drops in accuracy and F1 score. Conversely, setting $N_e = 16$ introduces excessive redundant entities and relations, which solely provide marginal utility for entity identification and may impose considerable burdens for the 7B MLLM.

The bottom sub-figures compare different visualization styles. Contrary to our initial hypothesis that entity bounding boxes would enhance MLLM's entity-relation discrimination, the box style underperforms the default plain style. Further experimentation with star-shaped borders yields even more pronounced performance deterioration, suggesting that explicit segmentation boundaries may disrupt the spatial reasoning of MLLMs. Typography and layout choices also significantly affect performance: Replacing "Times New Roman" with the cursive "Segoe Script" severely degrades results due to OCR difficulties. The SFDP layout (with straight-line edges) produces sparse graphs containing more edge crossings, resulting in inferior performance compared to the default layout. More sampled KG images can be found in Appendix D.3.

### 4.6 CASE STUDY

To evaluate the extent to which MLLMs can comprehend and reason over KG images, we design a case study involving a complex KG image comprising 27 entities and 41 triples, and reserve 40 additional triples related to these entities for the completion task.

The results are shown in Figure 5. We use Qwen2.5-VL-72B as the MLLM and task it with three objectives: listing all entities, listing all relations, and completing 40 triples in the form of *(head entity, relation, ?)*, where "?" denotes the missing entity. From the results we can observe that, the MLLM demonstrates remarkable capability in understanding the KG image even without fine-tuning, accurately identifying all entities and relations with just a single relation oversight. Furthermore, it performs exceptionally well in completing KG triples. All predicted triples are factually correct, though 9 triples were already present in the image, possibly due to hallucination. Notably, there are also 6 predicted triples absent from the dataset yet verified as correct through manual validation. As more advanced MLLMs continue to emerge, we anticipate these results may still have room for improvement. The details of case study are provided in Appendix D.4.

### 5 CONCLUSION AND LIMITATION

In this paper, we propose SeeKG, a novel MLLM-based framework for KG reasoning with visualized sub-KG images as input. SeeKG features an end-to-end pipeline integrating search, pruning, and visualization modules to efficiently sample relevant subgraphs and transform them into highly reliable, customizable graphical representations. Experimental results indicate that SeeKG outperforms all LLM-based baselines by substantial margins with a 7B MLLM as backbone. A potential limitation for our work is the unexplored integration of advanced LLM techniques, such as reinforcement learning from human feedback and chain-of-thought, which we reserve for future work.

ETHICS STATEMENT

This research strictly adheres to established scientific ethical guidelines. All datasets used in experiments are obtained from open-source repositories. The primary MLLMs referenced are also publicly available. Consequently, this study involves no collection of sensitive or personally identifiable data, nor does it raise any ethical concerns regarding data provenance or usage rights.

REPRODUCIBILITY STATEMENT

To ensure full reproducibility of our research, we have provided comprehensive methodological details and uploaded the complete source code. Additionally, we explicitly disabled temperature sampling in all experiments with LLMs to guarantee deterministic outputs and eliminate potential randomness in model responses.

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

| | |
|---|---|
| **Instruction** | Given the input knowledge graph triple in the form of (head, relation, tail), please first identify the key entities/relationships in the KG image, then cross-reference with your internal knowledge to determine the validity of the triple. Note: the image doesn't contain the input triple, and relation definitions may vary in strictness. Return True if the probability ≥ 0.5, otherwise False. Respond strictly with '**True**' or '**False**' only. Do not include any additional text. |
| **User Input** | The input triple: (**Veniamin Smekhov, country of citizenship, Soviet Union**)

The input knowledge graph image: **\<image\>** |
| **Response** | **True** |

Figure 6: The instruction template used in the main experiment for both training-free and supervised fine-tuning settings.

## A  RELATED WORKS TO CONVENTIONAL KG REPRESENTATION LEARNING

Conventional KG representation learning focuses on the encoding of relational structures via the triple-based methods or GNNs (Ji et al., 2020; Chen et al., 2025). The triple-based methods (Bordes et al., 2013; Wang et al., 2014; Lin et al., 2015a; Trouillon et al., 2016; Dettmers et al., 2018; Balazevic et al., 2019; Guo et al., 2019; Sun et al., 2019) design diverse score functions to evaluate a given triple $\tau = (e_i, r_k, e_j)$. For example, TransE (Bordes et al., 2013) models $\tau$ as $\mathbf{e}_i + \mathbf{r}_k = \mathbf{e}_j$, where the boldfaced denote the corresponding embeddings. The following methods such as ComplEx (Trouillon et al., 2016) and RotatE (Sun et al., 2019) further extend TransE to the complex space and polar coordinates. GNN-based methods (Schlichtkrull et al., 2018; Wang et al., 2018; Vashishth et al., 2020; Sun et al., 2020; Chen et al., 2021; Guo et al., 2022; Zhang et al., 2024a), such as CompGCN (Vashishth et al., 2020) and DAN (Guo et al., 2024b), usually follow a two-step paradigm: first aggregating the neighboring entities into embeddings via GNNs and then employing the triple-based score functions for relational learning.

## B  INSTRUCTION TEMPLATE

The instruction template used in the main experiment is shown in Figure 6, which follows a structured format:

- Instruction specifies the task: analyze input knowledge graph triple, then extract key entities/relations from the KG image and cross-reference with internal knowledge to assess triple validity, finally return True/False based on probability. We restrict response to only True or False to ensure a fair comparison with existing methods.
- User input provides the input triple and KG image, e.g., *(Veniamin Smekhov, country of citizenship, Soviet Union)* and the placeholder <image>.
- Response is the expected output, which is used for fine-tuning and evaluation.

This template ensures clear task definition, consistent input-output structure, and precise response constraints, enabling LLMs to focus on knowledge reasoning and validity assessment.

## C  DATASET DETAILS

As shown in Table C, we leverage two benchmark datasets for evaluating SeeKG:

- CoDeX-S (2,034 entities, 42 relations) balances scale with interpretability, derived from Wikipedia hyperlinks.
- FB15K-237N (13,104 entities, 93 relations) offers large-scale complexity, masking inverse relations to reduce redundancy.

| Dataset | $\|\mathcal{E}\|$ | $\|\mathcal{R}\|$ | # Train | # Valid(+/-) | # Test(+/-) |
|---|---|---|---|---|---|
| CoDeX-S (Safavi & Koutra, 2020) | 2,034 | 42 | 32,888 | 1,827/1,827 | 1,828/1,828 |
| FB15K-237N (Lv et al., 2022) | 13,104 | 93 | 87,282 | 7,041/7,041 | 8,226/8,226 |

Table 3: Statistical information of datasets. The positve (+) and negative (-) samples are 1:1 in the valid / test set.

| Datasets | LLM | LoRA r | LoRA dropout | LoRA target modules | train batch size per device | loss criterion | gradient accumulation steps | optimizer |
|---|---|---|---|---|---|---|---|---|
| CoDeX-S | Qwen2.5-VL-7B | 64 | 0.05 | all | 32 | CausalLMLoss | 1 | AdamW |
| FB15K-237N | Qwen2.5-VL-7B | 32 | 0.05 | all | 32 | CausalLMLoss | 1 | AdamW |

| | # epoch | warm up ratio | max gradient norm | training max samples | image max pixels | MLLM cutoff length | sub-KG max entities | simple path max length |
|---|---|---|---|---|---|---|---|---|
| CoDeX-S | 2.0 | 0.1 | 1.0 | 10,000 | 262,144 | 2,048 | 6 | 3 |
| FB15K-237N | 2.0 | 0.1 | 1.0 | 20,000 | 262,144 | 2,048 | 8 | 3 |

Table 4: Hyper-parameter settings for supervised fine-tuning in the main experiments.

All datasets enforce a 1:1 ratio of positive-to-negative samples in validation/test splits, ensuring unbiased metric computation. Training sets vary in size to accommodate different learning regimes.

# D    EXPERIMENT DETAILS

## D.1    HYPER-PARAMETER SETTINGS

The detailed hyper-parameter settings are shown in Table 4. We use Qwen2.5-VL-7B (Wu et al., 2025) as the backbone LLM in the main experiment, and develop the training and evaluation procedure using LLamaFactory (Zheng et al., 2024) and VLLM (Kwon et al., 2023). All experiments are conducted using a single H100 GPU. We employ low-rank adaption (LoRA) (Hu et al., 2021) in the supervised fine-tuning setting, where $r = 64$ is identical to the baseline method KoPA (Zhang et al., 2024b). The learning rate and number of training epoch are set to $0.0001$ and $2.0$, respectively.

## D.2    DETAILED RESULTS OF SEEKG WITH DIFFERENT BACKBONE MLLMS

We present the detailed results of SeeKG across various backbone MLLMs in Figure 7. Evidently, there are substantial performance variations in recall metrics among different MLLMs. Notably, while Gemini 2.5 achieves the highest precision in triple classification, its recall performance is the weakest among compared models, consequently yielding suboptimal accuracy and F1 scores, the similar to GPT-4o-mini. Even though, most MLLMs still have significantly better results than the single-modal LLMs in Table 1.

## D.3    MORE SAMPLED KG IMAGES WITH DIFFERENT PRUNING AND VISUALIZATION SETTINGS

We provide more KG image samples with different pruning and visualization settings in Figure 8. It is clear that including more entities inevitably expands the neighborhood of both source and target entities, as the number of simple paths is often limited. Also, the images with Star, Script and SFDP styles exhibit significant readability challenges.

## D.4    DETAILED RESULTS OF CASE STUDY

We present the details of case study in Figure 9, which includes the original input KG image, prompt instructions, and MLLM outputs. The sole missing relation in the output is "place of death", positioned at the right edge of the KG image.

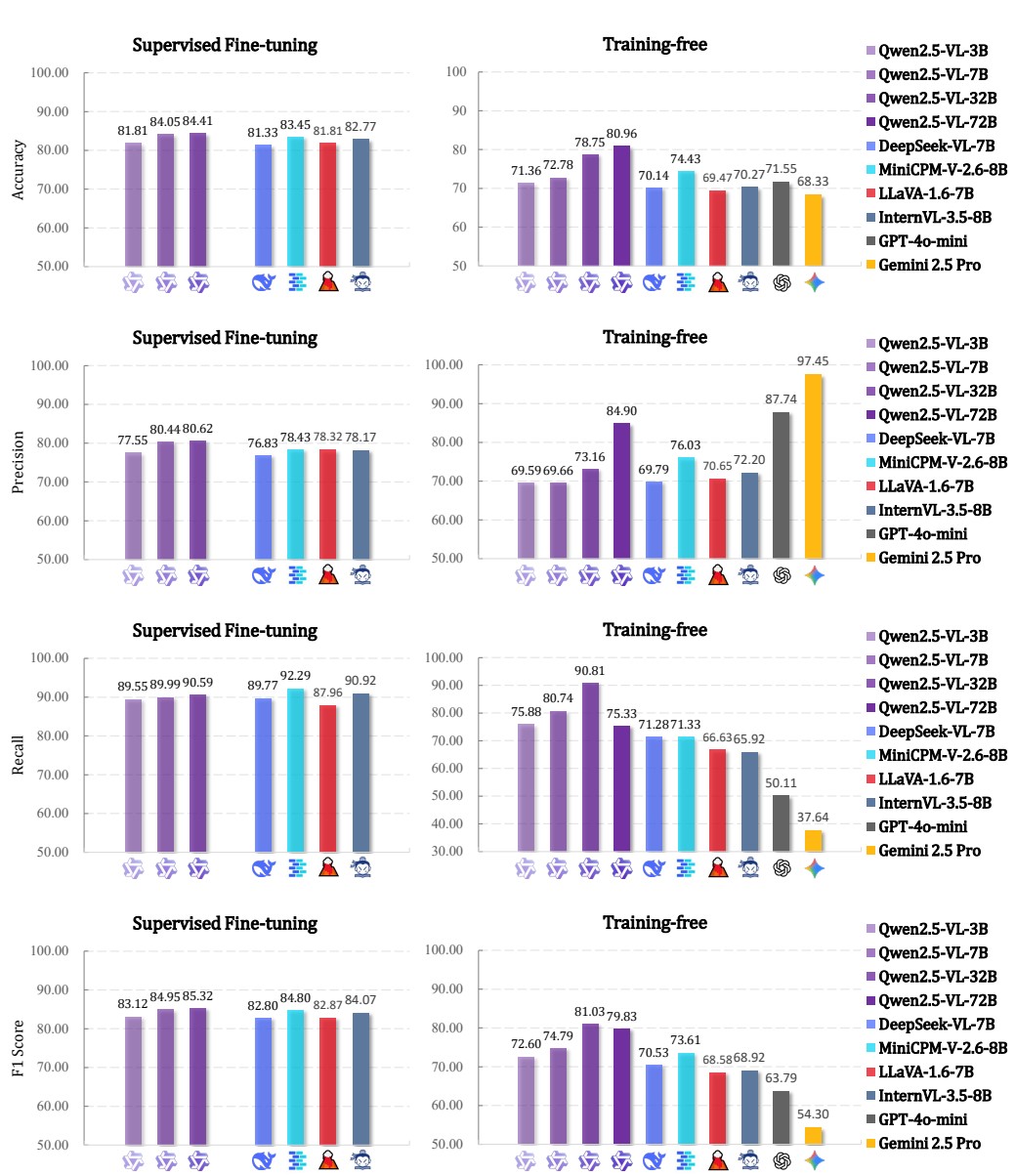

Figure 7: The detailed results of SeeKG with different backbone MLLMs on the CoDeX-S dataset.

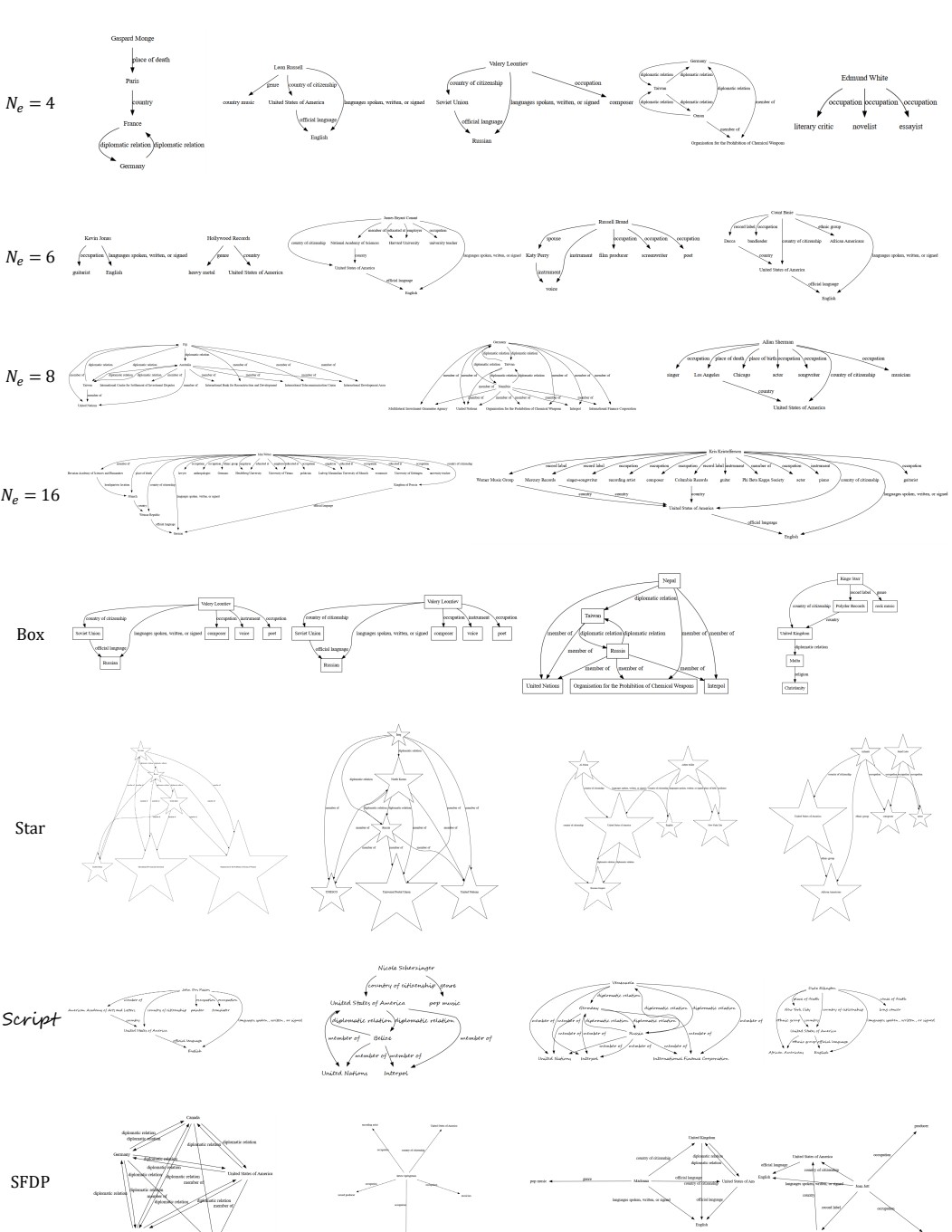

Figure 8: More sampled KG images with different pruning and visualization settings.

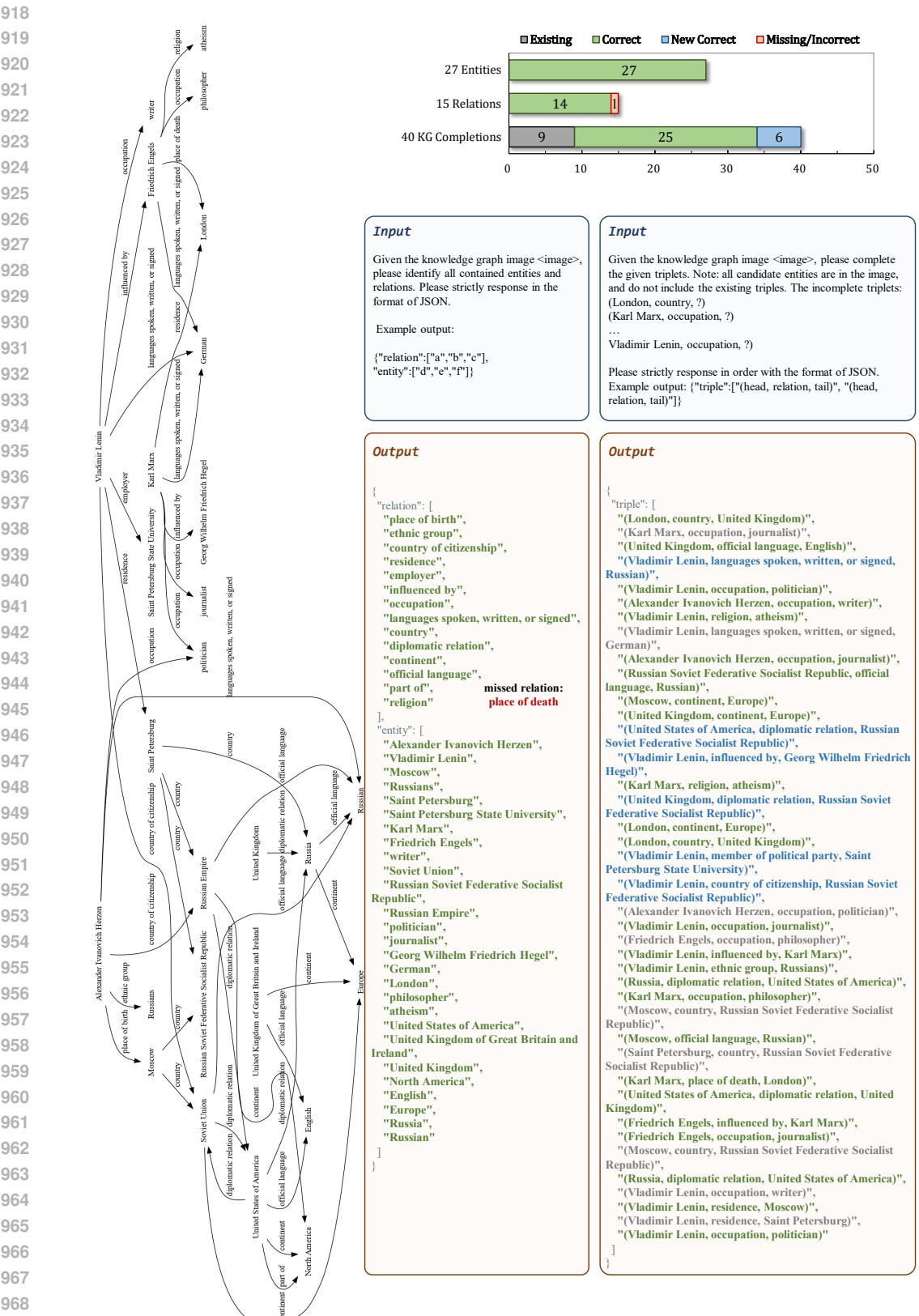

Figure 9: The detailed results of case study.

