# OpenReview forum: "Thinking is Seeing: Multi-modal Large Language Models are Exceptional in Understanding Knowledge Graphs"
_ICLR.cc/2026/Conference — ICLR 2026 Conference Withdrawn Submission_

### Official Review · Reviewer_QkkW · 2025-10-29

**Soundness:** 2
**Presentation:** 3
**Contribution:** 1
**Rating:** 2
**Confidence:** 5

**Summary:**

This paper propose to transform knowledge graphs into visually rendered representations as image, and then input into MLLM for leveraging advanced vision capabilities. Leveraging visual understanding of knowledge graphs is an important and promising direction because of the inherent visual symbols and connections within them.
The authors propose SeeKG, and carry out training-free and supervised fine-tuning settings to evaluate performance. They conduct experiments on two small datasets (CoDeX-S and FB15K-237N) and compare with LLM-based methods and traditional embedding-based methods.

**Strengths:**

--The direction explored in this paper is valuable.

--The ablation study and case study are sufficient.

**Weaknesses:**

Representation：

--Understanding KGs through MLLM is not a new direction; this paper lacks elaboration and comparison with existing work.

--The authors need to polish their paper writing and carefully declare their contributions.

Method：

--The technical contributions are limited. Specifically, rendering knowledge graphs into images, using training-free or SFT-based method to evaluate MLLM, are not novel.

--Compared to existing work, this paper does not contribute any new ideas or technologies.

Experiment：

--The experiments are insufficient. The authors conduct limited experiments on only two small synthetic datasets.

--The comparison is unreasonable. The authors should compare their work with similar methods in the same field to demonstrate its true contribution.

Some Reference:
[NeuIPS 2025] Gita: Graph to visual and textual integration for vision-language graph reasoning
[CVPR 2025] Mosaic of modalities: A comprehensive benchmark for multimodal graph learning

**Questions:**

Key questions:

--Distinctions from existing works;

--Novel perspectives offered in this paper compared to prior research;

--Comprehensive experiments, including justified dataset and baseline selections.

---

### Official Review · Reviewer_61Mg · 2025-10-30

**Soundness:** 2
**Presentation:** 1
**Contribution:** 2
**Rating:** 2
**Confidence:** 4

**Summary:**

This paper proposes SeeKG, an MLLM-based framework for knowledge graph (KG) reasoning that leverages visualized sub-KG images as input to address the triple classification task. The framework supports various MLLMs, flexible sub-KG sampling strategies, and customizable visualization settings. Experimental results on KG triple classification (Table 1), alongside ablation studies evaluating key components (Table 2), demonstrate the effectiveness of the proposed approach.

**Strengths:**

This paper explores the use of visualized knowledge graph images and MLLMs to tackle the triple classification task in KG completion. The results presented in Table 1 demonstrate that multi-modal models are capable of effectively leveraging visual KG representations for this task. In addition, the authors conduct a comparative evaluation across different families of vision-language models, including both in-house and open-source variants, to further validate the robustness of their proposed setting.

**Weaknesses:**

1. **Limited and unclear contribution**. The overall contribution of this paper remains unclear and appears limited. The scope is narrow, as the method only targets the triple classification task in KG completion. In addition, several claims are overstated or inaccurate. For example, the authors state that “_rather than adapting MLLMs for multi-modal KG tasks, we explore the correlation between the visual capacity of MLLMs with abstract thinking_” (line 141–142). However, the paper strictly evaluates models on triple classification, which primarily involves symbolic relational reasoning within a predefined knowledge graph. In contrast, abstract thinking generally requires higher-level conceptualization, analogical reasoning, and commonsense inference grounded in perceptual understanding. These capabilities that are not adequately assessed by the presented experiments. Therefore, this claim is not well supported by the evidence.

2. **Ambiguous and unfair comparisons in Table 1**. The comparison in Table 1 is problematic.
(1) Most baselines are LLM-based approaches, whereas SeeKG is multi-modal, yet this critical distinction is not clearly emphasized. The visual separation in the table is subtle and may lead readers to misinterpret SeeKG as an LLM-only method.
(2) The baselines utilize different-scale backbones: SSQR is based on LLaMA-2 7B, while SeeKG uses Qwen2.5-VL-7B, which introduces additional visual capabilities and a different LLM backbone (Qwen2.5 Transformer). The fairness of the comparison is thus questionable.
(3) In Table 2, the notation “w/” appears to be used incorrectly; it seems intended to indicate “w/o” but is currently ambiguous.

3. **Questionable results in Figure 3**. I am skeptical about the results reported in Figure 3. It is unexpected that the zero-shot performance of Gemini 2.5 Pro is reported lower than that of Qwen2.5-VL-3B. Given that Gemini 2.5 Pro typically exhibits superior multimodal reasoning ability, this counterintuitive outcome requires further explanation or ablation to ensure the correctness of the evaluation setup.

4. **Missing related work on graph-based multimodal reasoning**. The related work section is incomplete. Section 2 only discusses LLMs for KG reasoning, while recent studies have demonstrated strong LMM capability on visual-graph reasoning tasks. Relevant literature, such as VisionGraph [1], GITA [2], and GraphArena [3], should be discussed to better situate this work within the broader research landscape. Omitting these works weakens the novelty claim.

5. **Uninformative case study analysis (Figure 5)**. The case study provided in Figure 5 is unclear and does not offer meaningful insight. A useful case study should highlight why the proposed model succeeds (or fails), revealing strengths and limitations of the approach. However, the example merely reports an example of complicated KG image and model's results. This prevents readers from understanding how SeeKG reasons or where it may struggle.

6. **Unsupported claims regarding reasoning validity (Figure 1)**. Figure 1 showcases an example reasoning chain as output by an MLLM and claims that the model “sufficiently understands the graph context and integrates visual clues with its internal knowledge.” Yet, no experiments substantiate whether the intermediate reasoning content is reliable. Given that hallucinations remain a common issue in MLLMs, these claims require empirical validation. For example, the authors could analyze:
(1) the correctness of generated reasoning vs. ground-truth facts,
(2) cases where the reasoning text is incorrect but the final prediction is correct.

[1] Li, Yunxin, et al. "Visiongraph: Leveraging large multimodal models for graph theory problems in visual context." ICML 2024.
[2] Wei, Yanbin, et al. "Gita: Graph to visual and textual integration for vision-language graph reasoning." NeurIPS 2024.
[3] Tang, Jianheng, et al. "Grapharena: Evaluating and exploring large language models on graph computation." ICLR 2025.

**Questions:**

Please see the weaknesses part.

---

### Official Review · Reviewer_qWdC · 2025-11-05

**Soundness:** 2
**Presentation:** 3
**Contribution:** 2
**Rating:** 4
**Confidence:** 4

**Summary:**

This work propose SeeKG, which convert the KG into images (i.e., graph visualization), and wish to prove the benefits of such vision-based structure perception on KG completion. The experiments show it works on 2 datasets, however, some insights of its motivation and techniques it adopted are established, make its idea and technique novelty neural.

**Strengths:**

1. The writing is easy to understand
2. The motivation make sense, adopt this on KGC is valuable.
3. Results on 2 datasets are promising

**Weaknesses:**

Idea Novelty: While the motivation behind the paper: Leveraging graph visualization to enhance graph structural comprehension and benefit from visual perception, such as substructure sensitivity is sound, this concept is not particularly new. Prior works, such as [1] and [2], have explored similar directions. Specifically,  GITA [1] demonstrates that transforming graph structures into images and processing them through VLMs enhances fundamental graph-related tasks, highlighting the benefits of visualizing structures and substructures. Similarly, [2] investigates the efficacy of graph visualization in GNNs with dedicated experiments focusing on substructure sensitivity. Both studies substantially cover the core idea of this paper in terms of utilizing vision to better understand structure and substructure, addressing comparable problems, and employing similar tasks, such as link prediction, which overlaps conceptually with KGC. Another problem is that this paper avoid all these works in discussion. Therefore, the paper's contribution in terms of idea novelty appears somewhat overstated.

Technical Novelty: The proposed approach of integrating structural or visual inputs via special tokens is also not particularly novel, as it has been a well-established practice in numerous multimodal approaches. For instance, LLaVA [3] employs special tokens for vision input, and LLaGA [4] utilizes a similar mechanism to encode graph structure. Consequently, the technical design does not introduce distinctly novel methodologies.

Domain-Specific Contributions: While adapting existing insights and techniques to KGC is a meaningful contribution, the paper could benefit from demonstrating more distinctive designs or considerations tailored specifically to Knowledge Graph (KG) visualization, e.g., how perform current KG visualization versus other visualization approach, why such visualization, how benefits? Such domain-specific contributions should be central to works of this nature but are not strongly evident in the current approach (There are some, but not enough from my point of view).

Experimental Scope and Metrics: The experimental evaluation suffers from a limited scope, being conducted on only two datasets. To ensure robust results, it is important to test the approach on a broader range of datasets commonly used for KGC, such as WN18RR, UMLS, and Yelp. Additionally, the evaluation should encompass a more comprehensive set of standard metrics, including Mean Reciprocal Rank (MRR), Hits@1, Hits@3, Hits@10, and Hits@100, along with reporting standard deviations to provide a clearer picture of performance reliability. Besides, the baselines are not powerful, particular for the LLM-based training free ones, there have been many SOTA frameworks for KGC in such path. More necessary, time/memory analysis, OCR ability rely contents, are need to be exposed.

[1] GITA: Graph to Visual and Textual Integration for Vision-Language Graph Reasoning, NeurIPS 2024

[2] Open Your Eyes: Vision Enhances Message Passing Neural Networks in Link Prediction, ICML 2025

[3] Visual Instruction Tuning, NeurIPS 2023

[4] LLaGA: Large Language and Graph Assistant, ICML 2024

**Questions:**

1. What about the performance on small models 2B/1.5B for resource-limited setting?
2. The time/memory comparison with other methods?
3. How can such approach be extend to large KG, and how can current visualization handle the ambiguous/similar entities, like winner of  ICPC 2023 and winner of ICPC 2024, like The U.S. and America.  Is it depends on the OCR capabilities and how much? What if when the OCR failed or errored, like recognize the 2026 to 2025, e.g.
5. The semantic label is massive, how to make the KG clearly presented in the canvas when visualized?
6. The absent hyperparameters claim and sensitive studies?
7. For the comparison baseline of LLM-based train-free group, why it only contains trivial ICL/ZS, but not other SOTA LLM-based KGC training-free frameworks in recent years?

---

### Official Review · Reviewer_BHn7 · 2025-11-05

**Soundness:** 2
**Presentation:** 2
**Contribution:** 2
**Rating:** 4
**Confidence:** 4

**Summary:**

This paper proposes SeeKG, which transforms KGs into visually rendered representation as image inputs for MLLMs to facilitate conceptual thinking and abstract reasoning of MLLMs. The authors evaluate SeeKG under both training-free and supervised fine-tuning settings and demonstrate that it achieves better performance than existing methods that combine LLM with KG.

**Strengths:**

The idea of using visualized KG as the context of MLLM seems novel and interesting.

**Weaknesses:**

- Technical contribution is a bit limited. Generally the proposed method simply uses the visualized KG as the context of MLLM (possibly with some further fine-tuning).
- Performance improvements seem not significant enough to support the proposed method.

**Questions:**

- In Table 1, the classical baseline methods are restricted to embedding-based methods and ignore GNN-based methods like NBFNet (Zhu et al., 2021, mentioned in Section 1) or RED-GNN [1]. As these methods can often yield better performance than embedding-based methods, some discussion (preferably with some empirical comparison) should be necessary here.
- While the performance improvements in Table 1 are a bit incremental, the authors may need to consider the computational cost of their proposed method, including:
    - How does the computational cost scale with the number of parameters (e.g., from Qwen2.5-VL-3B, 7B to 32B, 72B)?
    - How does the fine-tuning method introduce additional computational cost than its training-free counterparts? Furthermore, given that embedding-based methods easily excels training-free SeeKG, how does such computational cost compare against these embedding-based methods?
- I am a bit uncertain if there is possible information leakage in the experiments, as the baseline without any context achieves only slightly worse performance than the complete proposed method in the ablation study. Some discussion should be welcome here, and the authors are encouraged to consider more specific KG (e.g., medical or scientific) to better avoid possible information leakage.
- Despite the GNN-based methods (NBFNet and RED-GNN), some other related works [2,3,4] may also worth some discussion to better support the superiority of proposed method.

## References
[1] Knowledge Graph Reasoning with Relational Digraph. WWW 2022

[2] Think-on-Graph: Deep and Responsible Reasoning of Large Language Model on Knowledge Graph. ICLR 2024

[3] Generate-on-Graph: Treat LLM as both Agent and KG for Incomplete Knowledge Graph Question Answering. EMNLP 2024

[4] Open Your Eyes: Vision Enhances Message Passing Neural Networks in Link Prediction. ICML 2025

---

### Note · Authors · 2025-11-14

I have read and agree with the venue's withdrawal policy on behalf of myself and my co-authors.